# Detection and Characterization of Enterovirus B73 from a Child in Brazil

**DOI:** 10.3390/v11010016

**Published:** 2018-12-28

**Authors:** Geovani de Oliveira Ribeiro, Adriana Luchs, Flávio Augusto de Pádua Milagres, Shirley Vasconcelos Komninakis, Danielle Elise Gill, Márcia Cristina Alves Brito Sayão Lobato, Rafael Brustulin, Rogério Togisaki das Chagas, Maria de Fátima Neves dos Santos Abrão, Cassia Vitória de Deus Alves Soares, Steven S. Witkin, Fabiola Villanova, Xutao Deng, Ester Cerdeira Sabino, Eric Delwart, Antonio Charlys da Costa, Élcio Leal

**Affiliations:** 1Institute of Biological Sciences, Federal University of Pará, Pará 66075-000, Brazil; geovanibiotec@gmail.com (G.d.O.R.); fevface@gmail.com (F.V.); 2Enteric Disease Laboratory, Virology Center, Adolfo Lutz Institute, São Paulo 01246-000, Brazil; driluchs@gmail.com; 3Secretary of Health of Tocantins, Tocantins 77453-000, Brazil; flaviomilagres@uft.edu.br (F.A.d.P.M); eu3rafael@gmail.com (R.B.); chagastogisaki@hotmail.com (R.T.d.C.); fatima_abrao@yahoo.com.br (M.d.F.N.d.S.A.); cassiavitoriaalves@gmail.com (C.V.d.D.A.S.); 4Institute of Biological Sciences, Federal University of Tocantins, Tocantins 77001-090, Brazil; 5Public Health Laboratory of Tocantins State (LACEN/TO), Tocantins 77016-330, Brazil; eumarciaalvesbrito@gmail.com; 6Postgraduate Program in Health Science, Faculty of Medicine of ABC, Santo André 09060-870, Brazil; skomninakis@yahoo.com.br; 7Retrovirology Laboratory, Federal University of São Paulo, São Paulo 04023-062, Brazil; 8Institute of Tropical Medicine, University of São Paulo, São Paulo 05403-000, Brazil; degill@g.clemson.edu (D.E.G.); switkin@med.cornell.edu (S.S.W.); sabinoec@gmail.com (E.C.S.); 9Department of Obstetrics and Gynecology, Weill Cornell Medicine, New York, NY 10065, USA; 10Blood Systems Research Institute, San Francisco, CA 94143, USA; xdeng@bloodsystems.org (X.D.); eric.delwart@ucsf.edu (E.D); 11Department Laboratory Medicine, University of California San Francisco, San Francisco, CA 94143, USA; 12LIM/46, Faculty of Medicine, University of São Paulo, São Paulo 01246-903, Brazil

**Keywords:** enterovirus B73, virome, picornavirus, pediatric virology, gastroenteritis, phylogenetics

## Abstract

Enterovirus B73 is a new member of the *Enterovirus B* species. First detected in the USA, it has been subsequently identified in China, India, Oman, and the Netherlands. In this study, we characterize the first B73 strain (named TO-127) to be detected in South America. TO-127 was obtained from a child with acute gastroenteritis living in a rural area in Northern Brazil. The subject was not infected with any known enteric pathogens such as norovirus, rotavirus, helminths, or enteric bacteria. Analysis of the nearly full-length TO-127 genome (6993 nt) indicated a 74–75% nucleotide similarity with EV-B73 strains from other countries. Evolutionary analysis suggests that B73 is endemic and widespread.

## 1. Introduction

Implementation of next-generation sequencing (NGS) to clinical and environmental samples has led to the identification of new species of the Picornaviridae family [1]. This family is currently divided into 29 genera whose natural hosts are vertebrates, including mammals and birds [1]. Enteroviruses, polioviruses, hepatoviruses, and aphthoviruses are the most exhaustively characterized animal pathogens. They have been associated with a wide spectrum of clinical manifestations, including undifferentiated fever, respiratory illness, aseptic meningitis, and acute flaccid paralysis (AFP) [2].

Enteroviruses (EV) are composed of 12 species—enteroviruses A–I and rhinoviruses A–C—that are subdivided into more than 100 serotypes. Only EV-A, EV-B, EV-C, and EV-D species are known to infect humans. The EV-B species is composed of approximately 60 serotypes, comprising all echoviruses (E), coxsackievirus B (CV-B), and EV-B69, B73-75, B77-88, B93, B97, B98, B100, B101, B106, B107, and B110 [2].

EV-B73 belongs to the *Enterovirus B* family, which was initially isolated in 1955 in California/USA from a fecal sample [3]. Despite there having been several studies related to enterovirus surveillance which have employed high-throughput nucleotide sequencing technologies, only three full-length genomes of EV-B73 have been published in the GenBank database. We now report the first B73 enterovirus to be detected in Brazil, which is named TO-127.

## 2. Materials and Methods

### 2.1. Patient Information

This work was part of surveillance program carried out in the state Tocantins, north Brazil, from 2010 to 2016. A total of 238 fecal specimens collected between 2010 and 2016 were screened for enteric pathogens (i.e., rotavirus and norovirus), bacteria (i.e., *Escherichia coli* and *Salmonella sp*.), endoparasites (i.e., *Giardia sp.*), and helminthes, using conventional culture techniques and commercial enzyme immunoassays. Subjects ranging from 3 to 14 years old were suffering from acute gastroenteritis at the time of sampling. Rotaviruses (*n* = 112), adenoviruses (*n* = 44), norovirus (*n* = 39), astroviruses (*n* = 8), and sapovirus (*n* = 8) were identified in some of the subjects. In 2015, fecal specimens from a boy (born in Araguaina City in 2013) presenting a fever (40 °C) and affected by acute gastroenteritis were subjected to screening for the main enteric pathogens (rotavirus, norovirus, *Escherichia coli*, *Salmonella spp*., *Giardia sp*., and helminths) using traditional detection methods (i.e., cultured techniques and commercial enzyme immunoassays). No known enteric pathogens were detected. To identity possible undetected enteric viruses, NGS techniques were applied using the method described below.

### 2.2. Sample Processing

The procedure used to perform deep-sequencing was a combination of several protocols normally applied to viral metagenomics and virus discovery, which have been partially described by da Costa et al. [4]. Briefly, 50 mg of the human TO-127 fecal sample was diluted in 500 μL of Hanks’ buffered salt solution (HBSS), added to a 2 mL impact-resistant tube containing lysing matrix C (MP Biomedicals, Santa Ana, CA, USA) and homogenized in a FastPrep-24 5G Homogenizer (MP biomedicals, USA). The homogenized sample was centrifuged at 12,000× *g* for 10 min, and approximately 300 μL of the supernatant was then percolated through a 0.45 μm filter (Merck Millipore, Billerica, MA, USA) to remove eukaryotic- and bacterial-cell-sized particles. Approximately 100 μL, roughly equivalent to one fourth of the volume of the tube, of cold PEG-it Virus Precipitation Solution (System Biosciences, Palo Alto, CA, USA) was added to the filtrate, and the contents of the tube were gently mixed and then incubated at 4 °C for 24 h. After the incubation period, the mixture was centrifuged at 10,000× *g* for 30 min at 4 °C. Following centrifugation, the supernatant (~350 μL) was discarded. The pellet, rich in viral particles, was treated with a combination of nuclease enzymes (TURBO DNase and RNase Cocktail Enzyme Mix-Thermo Fischer Scientific, Waltham, MA, USA; Baseline-ZERO DNase-Epicentre, Madison, WI, USA; Benzonase-Darmstadt, Darmstadt, Germany; and RQ1 RNase-Free DNase and RNase A Solution-Promega, Madison, WI, USA) to digest unprotected nucleic acids. The resulting mixture was subsequently incubated at 37 °C for 2 h.

After incubation, viral nucleic acids were extracted using a ZR & ZR-96 Viral DNA/RNA Kit (Zymo Research, Irvine, CA, USA) according to the manufacturer’s instructions. The cDNA synthesis was performed with an AMV reverse transcription reagent (Promega, WI, USA). Second strand cDNA synthesis was performed using a DNA Polymerase I Large (Klenow) Fragment (Promega, WI, USA). Subsequently, a Nextera XT Sample Preparation Kit (Illumina, San Diego, CA, USA) was used to construct a DNA library, which was identified using dual barcodes. For the size range, Pippin Prep (Sage Science, Inc., Beverly, MA, USA) was used to select a 300 bp insert (range 200–400 bp). The library was deep-sequenced using a Hi-Seq 2500 Sequencer (Illumina, CA, USA) with 126 bp ends. Bioinformatics analysis was performed according to the method previously described by Deng et al. [5]. The contigs, including sequences of rotaviruses as well as enteric viruses, humans, fungi, bacteria, and others, sharing a nucleotide identity of ≤95%, were assembled from obtained sequence reads by de novo assembly. The resulting singlets and contigs were analyzed using BLASTx to search for similarities to viral proteins in GenBank’s Virus RefSeq. The contigs were checked against the GenBank nonredundant nucleotide and protein databases (BLASTn and BLASTx). After identification of an enterovirus virus, a reference template sequence was used for mapping the full-length genome with Geneious R9 software (Biomatters Ltd. L2, Auckland, New Zealand).

### 2.3. Typing and Genomic Analysis

BLASTn was initially used to identify viral sequences via their sequence similarity to annotated viral genomes in GenBank. Based on the best hits of BLASTx searches, the following 23 genomes, listed by their GenBank numbers, were chosen to be used in the phylogeny inference: AY421760, M16560, B5L070215, MF962897, MF678308, MF678341, AF114384, AF083069, AF465517, AY302558, MF838733, KJ957190, KX774483, KX139459, KX139460, AY302552, NC030454, KF874626, AF504533, AF241359, AY843301, AY426531, and D00820. The TO-127 strain has been deposited in GenBank under the accession number MK069966.

The VP1 coding region of the TO-127 strain was analyzed using online Enterovirus Genotype Tool [6]. Phylogenetic trees using other EV-B73 strains were also constructed to classify (genotype) our B73 isolate. GenBank accession numbers of EV-B73 strains sequences used for phylogenetic comparison were AF504535, LC167442, LC120906, AF504537, AB474181, AB474182, JF718571, AF504533, GQ329839, AY919467, AY919524, AY919533, AY919535, KY866638, KY866640, KY866641, KY866642, KY866643, KX580675, AF241359, AF241363, AF241361, HQ538452, HQ538453, JN204049, JN204050, KF413011, AF241362, AB426614, and AB426613.

Additionally, genomic regions of the TO-127 strain were compared to three EV-B73 and three non-EV-B73 strains by pairwise alignment using the identity matrix tool of BioEdit 7.0.5.3 software [7].

### 2.4. Phylogenetic Analysis

A multiple sequence alignment was performed using Mafft software [8]. Subsequently, phylogenetic trees were constructed using MEGA software, version X [9] by a neighbor joining approach with a Kimura two-parameter model. Branch support values were assessed using the bootstrapping test with 1000 replications.

The estimated mean evolutionary divergence of sequences analyzed within and between EV-B73 clusters was performed using MEGA software, version X [9]. The analysis was performed using the Kimura 2-parameter replacement model with 1000 replicates of bootstrapping.

## 3. Results

The near full-length genome of a B73 enterovirus strain, named TO-127, was sequenced from a fecal sample using the NGS method. Its genomic characterization and phylogenetic relationships with other EVs are described below.

### 3.1. Typing and Genomic Analysis

Molecular typing of the VP1 region by an online tool [6] indicated that our isolate was an EV-B73 strain. Its nearly complete genome consisted of 6993 nt, which encoded a polyprotein with 2188 amino acids. The 5′ and 3′ ends of TO-127 consisted of 327 nucleotides. Since the 5′-UTR region of TO-127 was partially sequenced the genome is shorter than that of EV-B73 references (i.e., 088/SD/CNH/04 and CA55-1988). In addition, we performed comparisons to determine the similarity of TO-127 to other human enterovirus strains. Table 1 summarizes the similarity of TO-127 to EV-B73 strains 088/SD/CNH/04 and CA55-1988, and to enterovirus strains CA76-10392 and JV-10. Overall, the genome similarity of TO-127 to 088/SD/CNH/04 and JV-10 was 75% and 71%, respectively.

### 3.2. Phylogenetic Analysis

The near-full length genome (6993 nt) of TO-127 compared to 23 previously described EV-B virus genome strains is shown in Figure 1a. In this tree the Brazilian strain TO-127 clustered with other EV-B73 strains and the clade containing B73 strains had high branch support. This is consistent with preliminary molecular typing and similarity analysis. In general, strains of the same serogroup tended to cluster together (for example, E25, CVB5, and E6). It is important to note that strain CA55-1988 is not within the B73 clade because this isolate is a recombinant of the genomes of EV-B73—from which the coding region for structural proteins was derived—and the coxsackievirus B3 sequences of the non-structural region P2 [10].

Next, ten sequences corresponding to the partial VP1 region (730 nt) of EV-B73 were used to infer a phylogenetic tree. The phylogeny was divided into two clusters, with a high value of probabilistic support (~100%) (Figure 1b). Cluster 1 was composed of the strains isolated in India, China, the USA, and Brazil. Cluster 2 included strains from China, Oman, and the USA.

The average genetic diversity was estimated within and between each cluster. The mean diversity in Cluster 1 was 0.218 ± 0.01 and in Cluster 2 was 0.210 ± 0.01. The genetic diversity between Clusters 1 and 2 was 0.350 ± 0.05. Detailed analysis of recombination was performed in the polyprotein of the B73 TO-127 strain. Only the polyprotein region was used to accomplish this because most reference sequences of enteroviruses presented large indels in the 5′ and 3′ untranslated regions. We found one breakpoint in the position 2885, which corresponded approximately to the end of the VP1 gene, in the genome of the Brazilian strain TO-127 (Appendix A). Based on the position of the breakpoint we made two partitions of alignment and used each of them to construct a phylogenetic tree. The strain TO-127 has a discordant position in each tree, thus confirming the mosaic genome of B73 isolated in Brazil.

## 4. Discussion

The EV-B73 enterovirus serotype has been reported only infrequently. Although extensive studies on enteroviruses have been made, serotype B73 has been long considered untypeable by the traditional method of serological identification [3]; only after development of new molecular typing methods could the VP1 region and other serotypes be identified or re-classified [11]. Due to the limited number of worldwide EV-B73 isolates, epidemiological characteristics of this virus are scarce, as are identifications of its biological and pathogenic properties.

To date, nucleotide sequences from only 31 EV-B73 strains have been deposited in the GenBank database, of which 21 were isolated in Asiatic countries (China (*n* = 9), Bangladesh (*n* = 5), India (*n* = 3), Japan (*n* = 2), and Oman (*n* = 2)) and ten from other continents (USA (*n* = 4), the Netherlands (*n* = 5) and Nigeria (*n* = 1)). Some strains of EV-B73 have been detected from fecal samples of individuals with acute flaccid paralysis and acute gastrointestinal symptoms, but no other supplementary data has been able to be collected to conclude that EV-B73 was the cause of these illnesses. Except for its potential association with acute flaccid paralysis and gastrointestinal symptoms, the pathogenic profile of EV-B73 is still unknown. It should be noted that infections with enteroviruses are mostly asymptomatic, as has already been observed in EV-B73 studies [12,13,14]. Therefore, extensive surveys are needed to provide more conclusive information about the epidemiological and pathogenic profile of EV-B73. The present communication provides a molecular characterization of the nearly full-length genome of the first EV-B73 strain detected in South America. A genome tree has confirmed the relationship of the TO-127 strain to other EV-B isolates. All B73 strains cluster together, with the exception of CA55-1988 which is a recombinant strain [10]. The VP1 tree showed that B73 strains formed two distinct clusters, as has been reported by Liu [15]. The VP1 tree also indicated that TO-127 B is at the base of Cluster 1 and is closely related to the strain CA54-4454 identified in the USA. Another important feature revealed by VP1 tree analysis is the high genetic divergence (35%) between Cluster 1 and Cluster 2 strains. Taking into account this high level of divergence and the fact that there are strains from distinct regions of the world in each cluster, it is reasonable to conclude that B73 is ancient and widespread. Furthermore, our isolate TO-127 is a recombinant strain with a chimeric genome different from other B73 recombinants isolated elsewhere.

In conclusion, this is the first report of a serotype B73 enterovirus located in Brazil. Although B73 was detected in a child with severe gastroenteritis in whom no other enteric pathogen was detected, it is still not possible to definitively conclude that this virus caused the disease. However, our finding reinforces the need for expanded EV surveillance in diverse world regions to better evaluate the potential role of B73 in gastroenteritis, acute flaccid paralysis, and other diseases.

## Figures and Tables

**Figure 1 viruses-11-00016-f001:**
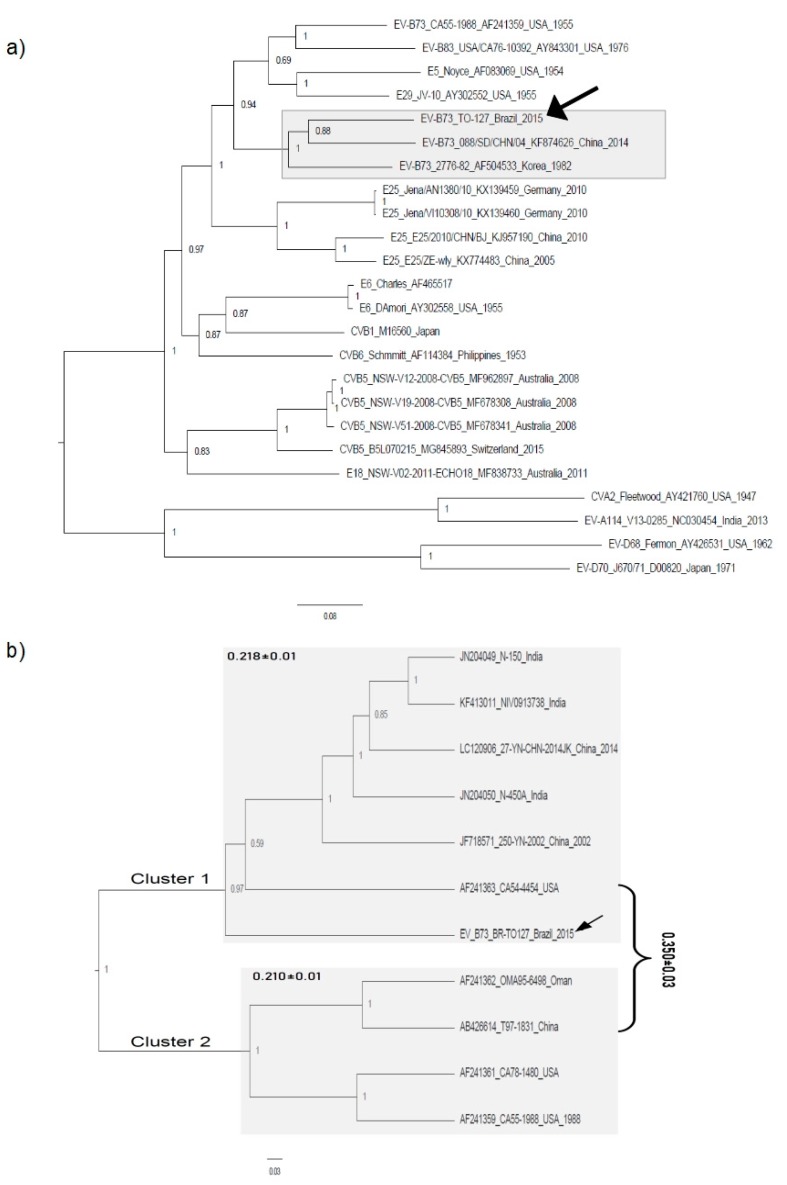
Phylogenetic trees of enterovirus B73. A phylogenetic tree based on the nearly complete genome constructed using sequences based on the best hits of BLASTn searches is shown in (**a**). A phylogenetic tree based on the VP1 region constructed using partial sequences of the coding region of the *vp1* gene of B73 isolates in shown in (**b**). The Brazilian strain TO-127 is indicated by filled arrows and the clade it is located in is indicated by gray areas. The vertical scale-bar colors indicate the branch support represented by the approximate likelihood ratio test (aLRT) and the scale bar under the tree represents the nucleotide substitution per site.

**Table 1 viruses-11-00016-t001:** Genome similarity (%) of TO-127 to EV-B73 and EV-B strains.

Region	Position ^1^	TO-127 versus EV-B73 ^2^		TO-127 versus EV-B ^3^	
		Nucleotide	Amino Acid	Nucleotide	Amino Acid
Partial 5′ UTR	1–327	76–82	Non coding	81–83	Non coding
VP4	328–528	72–78	82–95	70–77	81–87
VP2	529–1311	77–81	91–97	66–72	76–86
VP3	1312–2022	76–82	93–98	66–71	74–81
VP1	2023–2889	76–82	89–96	58–65	61–70
2A	2890–3339	72–78	87–93	78–80	92–95
2B	3340–3636	69–79	82–95	79–82	83–98
2C	3637–4623	79–82	95–97	80–82	97–98
3A	4624–4890	73–78	94–95	77–78	93–95
3B	4891–4956	72–84	90–95	80–85	94–95
3C	4957–5505	79–82	95–97	81–82	96–97
3D	5506–6891	76–81	90–97	79–82	96–97
3’ UTR	6895–6993	88–89	Non coding	88–91	Non coding
CDS	328–6891	79–80	94–96	74–76	87–90
Whole Genome	01–6993	79–81	Non coding	75–77	Non coding

^1^ Positions are numbered according to strain TO-127. ^2^ EV-B73 strains utilized for comparison and GenBank access number (088/SD/CNH/04 (KF874626) and CA55-1988 (AF241359)). ^3^ EV-B prototype strains utilized for comparison and GenBank access number (CA76-10392 (AY843301) and JV-10 (AY302552.1)).

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
