# Peer review of "Detection and Characterization of Enterovirus B73 from a Child in Brazil"

_viruses, 2018, doi:10.3390/v11010016_

Reviewer 1 Report

This article reports the identification of enterovirus B73 for the first time in Brazil from a boy with acute gastroenteritis symptoms. Enterovirus B73 has been identified only rarely in various countries. This highlights the importance of enterovirus surveillance for detection of rarely reported Human Enteroviruses serotypes, thus allowing a better understanding of the roles they play in disease. However, several points should be addressed by the authors to strengthen their conclusions.

1.    The selection of the case is not exactly clear. The authors state that human TO-127 fecal sample was obtained from a survey conducted from 2010-2016. However, it should be mentioned in detail how the selection of the fecal samples for NGS was made in the study subject. The authors note that " following an initial screening for enteric pathogens, NGS techniques were used to search for possible undetected enteric viruses". The authors have thus to give some additional and original data concerning the enteroviruses detected. For example: Is it possible to detect this virus in samples obtained from asymptomatic people? Were there other enteroviruses detected during a 6 year (2010-2016) period in the northern state of Tocantis, Brazil? What is the frequency of detecting Enterovirus B73 among enterovirus -positive subjects?

 2.    The authors report that a B73 strain of enterovirus was isolated from five-years old patient. It should be mentioned how the isolation of enterovirus B73 infection was made in the study subject. Please state the cell lines (i.e.  RD, Hep 2, etc) that were used to isolate the virus and passage numbers used in these studies. Cell sensitivity for enterovirus is likely to be a major determinant of the frequency with which it is reported. Was the Enterovirus 73 isolation confirmed by VP1 RT-PCR or was it based on NGS only?  Why is Enterovirus B73 isolate not used for typing and genomic analysis by NGS as an alternative or additional sample to fecal sample.

 3.    The patient information mentions that… the B73 enterovirus was isolated from a boy (born in 2013) and that was affected by acute gastroenteritis in 2015 at the clinical trial. The age provided in this section (two- years old patient) does not match the age on the Results section (lines. 138-139…a B73 strain of enterovirus was isolated from five-years old patient who presented with acute gastroenteritis in 2015).

 4.    The authors report no signal of recombination in the genome of TO-127. This is contrary to the general notion that genomic recombination is frequently seen in Enterovirus strains. Indeed, previous studies have shown intertypic recombination of Enterovirus B73 strains with other Enterovirus B viruses in the non-structural protein region (Liu, X.; Tao, Z.; Wang, H.; Lin, X.; Song, L.; Li, Y.; Zhang, L.; Wang, S.; Cui, N.; Xu, A. 278 Complete genome analysis of human enterovirus B73 isolated from an acute flaccid paralysis 279 patient in Shandong, China. Virus Genes 2014, 49, 38–44, doi:10.1007/s11262-014-1077-5). Surely the authors should provide an explanation for this.

Author Response

Reviewer 1

This article reports the identification of enterovirus B73 for the first time in Brazil from a boy with acute gastroenteritis symptoms. Enterovirus B73 has been identified only rarely in various countries. This highlights the importance of enterovirus surveillance for detection of rarely reported Human Enteroviruses serotypes, thus allowing a better understanding of the roles they play in disease. However, several points should be addressed by the authors to strengthen their conclusions.

1.    The selection of the case is not exactly clear. The authors state that human TO-127 fecal sample was obtained from a survey conducted from 2010-2016. However, it should be mentioned in detail how the selection of the fecal samples for NGS was made in the study subject. The authors note that " following an initial screening for enteric pathogens, NGS techniques were used to search for possible undetected enteric viruses". The authors have thus to give some additional and original data concerning the enteroviruses detected. For example: Is it possible to detect this virus in samples obtained from asymptomatic people? Were there other enteroviruses detected during a 6 year (2010-2016) period in the northern state of Tocantis, Brazil? What is the frequency of detecting Enterovirus B73 among enterovirus -positive subjects?

Resp: We appreciate this comment and understand the concerns about the occurrence of B73 in the general population. Our samples were limited to patients affected by gastroenteritis with no common agent associated.  We have included the following text to the section 2.1 Patient information : “A total of 238 fecal specimens collected between 2010 and 2016 were screened for enteric pathogens (i.e., rotavirus and norovirus), bacteria (i.e., Escherichia. coli and Salmonella sp.), endoparasites (i.e., Giardia sp.) and helminthes using conventional culture techniques and commercial enzyme immunoassays. Subjects ranging from 3 to 14 years old were suffering from acute gastroenteritis at the time of sampling. Rotaviruses (n=112), adenoviruses (n=44), norovirus (n=39), astroviruses (n=8) and sapovirus (n=8) were identified in some of the subjects.”

Since this is the first detection of B73 in Brazil there is no further information about the prevalence of this virus in healthy individuals.

The child whose B73 was isolated, no other pathogen (e.g. Salmonella,Escherichia, Norovirus, Rotavirus, helminthes) was detected.

2.    The authors report that a B73 strain of enterovirus was isolated from five-years old patient. It should be mentioned how the isolation of enterovirus B73 infection was made in the study subject. Please state the cell lines (i.e.  RD, Hep 2, etc) that were used to isolate the virus and passage numbers used in these studies. Cell sensitivity for enterovirus is likely to be a major determinant of the frequency with which it is reported. Was the Enterovirus 73 isolation confirmed by VP1 RT-PCR or was it based on NGS only?  Why is Enterovirus B73 isolate not used for typing and genomic analysis by NGS as an alternative or additional sample to fecal sample.

Resp: We are sorry that this text is not clear. I did not perform any cell culture nor PCR reaction to detect B73 in the fecal sample. Samples were prepared (the full description of this technique can be found in the reference 4) then Next Generation sequencing was used to detect DNA.  We also changed “isolated”to “detected” in the manuscript.

3.    The patient information mentions that… the B73 enterovirus was isolated from a boy (born in 2013) and that was affected by acute gastroenteritis in 2015 at the clinical trial. The age provided in this section (two- years old patient) does not match the age on the Results section (lines. 138-139…a B73 strain of enterovirus was isolated from five-years old patient who presented with acute gastroenteritis in 2015).

Resp: Our mistake. The patient was born in 2013 and in 2015 this child was affected by gastroenteritis. We corrected the text accordingly.

The authors report no signal of recombination in the genome of TO-127. This is contrary to the general notion that genomic recombination is frequently seen in Enterovirus strains. Indeed, previous studies have shown intertypic recombination of Enterovirus B73 strains with other Enterovirus B viruses in the non-structural protein region (Liu, X.; Tao, Z.; Wang, H.; Lin, X.; Song, L.; Li, Y.; Zhang, L.; Wang, S.; Cui, N.; Xu, A. 278 Complete genome analysis of human enterovirus B73 isolated from an acute flaccid paralysis 279 patient in Shandong, China. Virus Genes 2014, 49, 38–44, doi:10.1007/s11262-014-1077-5). Surely the authors should provide an explanation for this.

Resp: In the original manuscript we used simplot software (bootscan method) and no recombinant was detected in the strain TO-127 and strain. This time we used software RDP v.4 [http://darwin.uvigo.es/rdp/rdp.html), which utilizes a collection of methods and results were summarized in figure S1.

We found one breakpoint in the end of VP1 gene in TO-27. The position of this breakpoint was used to generate two partitions in the alignment: segment A (nucleotides 1 to 2959) and segment B (nucleotides 2960 to 3148 ). These partitions were used to construct phylogenetic tree that showed discordant locations of TO-127 between these trees. TO-127 clustered with the clade formed by B73 isolates in the tree of segment A and clustered in the clade formed by B5 strain in the tree constructed with the segment B. It is premature to say that TO-127 is a B73/B5 chimera because there are few reference “pure”strains available.

Reviewer 2 Report

This manuscript presents a nearly complete sequence of an enterovirus B73, a rarely isolated enterovirus B.  Only two complete sequences are known, and this is the only one from Brazil (most are Asian).  This report is more than a sequence announcement as phylogenetic analysis is done but other enterovirus B73 sequences have been reported. The finding that the original reported sequence was of a strain which was recombinant with coxsackievirus B3 and that there are two genotype clusters of enterovirus B73 was reported earlier (Liu et al 2014).  The unique findings of this report is of a South American enterovirus B73 and that it conforms to previous findings.  This lowers the significance of this brief report.

On lines 143-147, the manuscript refers to the complete genome being shorter and with a shorter 5’ UTR than the two complete genomes previously defined.  As this lack of 5’ sequence is undoubtedly due to a less than full length sequence, this text should be changed.  I do note that on line 158, the genome is noted to be near full length. I expect the authors did not intend to imply that their strain has a 327 nt 5’ UTR as a lack of 5’ terminal sequences is fairly frequent in attempts to sequence directly from environmental samples as in the nearly complete clone of EVB73 strain 2776-82 (Norder et al. 2002).  For this reason an alignment of this Brazilian strain sequence in the 5’ UTR should be utilized to define what portion of the complete 5’ UTRs in Genbank (088/SD/CNH/04 and CA55-1988) aligns with this sequence.  This then can be used in Table 1 to define the actual identity of the 5’ UTR sequence with the portion of the complete sequence contained by MK069966 (the TO-127 strain sequence).  I expect the genome similiarity for this portion of the 5’ UTR would be much greater than 35-39% or 25-35% as this is a misleading figure based on having a partial 5’ UTR sequence. The same is true for the line in Table I which compares the complete sequence.  I apologize if the GenBank entry for the TO-127 sequence addresses this by referring by defining the partial sequences but I was not given access to this GenBank entry.

Minor notes:

Lines 50 and 55: 

Reference 2 is an incorrect reference to a review of the Parvovirus family.  A better reference for this might be:

M. Pallansch SO, L.J. Whitton. Enteroviruses: polioviruses, coxsackieviruses, echoviruses, and newer enteroviruses. In: D.M. Knipe PMH, ed. Fields' Virology. 6th edition ed. Philadelphia, PA 19106, USA: Lippincott Williams & Wilkins, 2013.

Line 92-93: What primer was used for the RT reaction, random primers or oligo d(T)?

Author Response

Reviewer 2

This manuscript presents a nearly complete sequence of an enterovirus B73, a rarely isolated enterovirus B.  Only two complete sequences are known, and this is the only one from Brazil (most are Asian).  This report is more than a sequence announcement as phylogenetic analysis is done but other enterovirus B73 sequences have been reported. The finding that the original reported sequence was of a strain which was recombinant with coxsackievirus B3 and that there are two genotype clusters of enterovirus B73 was reported earlier (Liu et al 2014).  The unique findings of this report is of a South American enterovirus B73 and that it conforms to previous findings.  This lowers the significance of this brief report.

Resp: Although this study is descriptive it is important to characterize rare viruses in order to get better understanding the biology and potential risks of B73 to humans.

On lines 143-147, the manuscript refers to the complete genome being shorter and with a shorter 5’ UTR than the two complete genomes previously defined.  As this lack of 5’ sequence is undoubtedly due to a less than full length sequence, this text should be changed.  I do note that on line 158, the genome is noted to be near full length. I expect the authors did not intend to imply that their strain has a 327 nt 5’ UTR as a lack of 5’ terminal sequences is fairly frequent in attempts to sequence directly from environmental samples as in the nearly complete clone of EVB73 strain 2776-82 (Norder et al. 2002).  For this reason an alignment of this Brazilian strain sequence in the 5’ UTR should be utilized to define what portion of the complete 5’ UTRs in Genbank (088/SD/CNH/04 and CA55-1988) aligns with this sequence.  This then can be used in Table 1 to define the actual identity of the 5’ UTR sequence with the portion of the complete sequence contained by MK069966 (the TO-127 strain sequence).  I expect the genome similiarity for this portion of the 5’ UTR would be much greater than 35-39% or 25-35% as this is a misleading figure based on having a partial 5’ UTR sequence. The same is true for the line in Table I which compares the complete sequence.  I apologize if the GenBank entry for the TO-127 sequence addresses this by referring by defining the partial sequences but I was not given access to this GenBank entry.

Resp: We appreciate this comments and changed the table 1 and also the text after the new analysis of the similarities in the 5'-UTR region. The GenBank entry is pending and not yet available. We also changed the description of 5'-UTR in the GenBank file.

Minor notes:

Lines 50 and 55:

Reference 2 is an incorrect reference to a review of the Parvovirus family.  A better reference for this might be:

M. Pallansch SO, L.J. Whitton. Enteroviruses: polioviruses, coxsackieviruses, echoviruses, and newer enteroviruses. In: D.M. Knipe PMH, ed. Fields' Virology. 6th edition ed. Philadelphia, PA 19106, USA: Lippincott Williams & Wilkins, 2013.

Resp: We appreciate commentary and corrected the reference.

Line 92-93: What primer was used for the RT reaction, random primers or oligo d(T)?

Resp: The cDNA synthesis is performed using both OligodT and random decamers primers

Reviewer 3 Report

This is an incredibly well-written paper describing the detection and characterization of detection and genome sequencing of enterovirus B73 from a patient in Brazil. I only have very minor comments.

The detection of the viral genome does not demonstrate infection, so the authors should revise the title and language that indicate infection was present in order to be precise.

Line 98/99 "paired end reads" instead of "ends" (or if not paired, specify "single end reads")

Line 198 - two commas are present after "Therefore"

Line 200 - it looks like there is no space after the period

Instead of "genome tree" and "VP1 tree", I would say something like, "phylogenetic tree based on the nearly complete genome sequences" or "phylogenetic tree based on the VP1 sequence"

Very nice paper!

Author Response

Reviewer 3

This is an incredibly well-written paper describing the detection and characterization of detection and genome sequencing of enterovirus B73 from a patient in Brazil. I only have very minor comments.

The detection of the viral genome does not demonstrate infection, so the authors should revise the title and language that indicate infection was present in order to be precise.

Resp: We deeply appreciate this comments. We have changed the title and many parts of the text to to better clarify the reading.

Line 98/99 "paired end reads" instead of "ends" (or if not paired, specify "single end reads")

Line 198 - two commas are present after "Therefore"

Line 200 - it looks like there is no space after the period

Instead of "genome tree" and "VP1 tree", I would say something like, "phylogenetic tree based on the nearly complete genome sequences" or "phylogenetic tree based on the VP1 sequence"

Very nice paper!

Round  2

Reviewer 1 Report

My concerns from my previous review have been addressed.